# High Glycolytic Activity Enhances Stem Cell Reprogramming of Fahd1-KO Mouse Embryonic Fibroblasts

**DOI:** 10.3390/cells10082040

**Published:** 2021-08-10

**Authors:** Ahmad Salti, Solmaz Etemad, Marta Suarez Cubero, Eva Albertini, Beata Kovacs-Szalka, Max Holzknecht, Elia Cappuccio, Maria Cavinato, Frank Edenhofer, Pidder Jansen Dürr

**Affiliations:** 1Institute of Molecular Biology and CMBI, Department of Genomics, Stem Cell Biology and Regenerative Medicine, University of Innsbruck, 6020 Innsbruck, Austria; Marta.Suarez-Cubero@uibk.ac.at (M.S.C.); Frank.Edenhofer@uibk.ac.at (F.E.); 2Center for Medical Research, University Clinic for Ophthalmology and Optometry, Johannes Kepler University Linz, 4020 Linz, Austria; 3Institute for Biomedical Aging Research, University of Innsbruck, 6020 Innsbruck, Austria; Solmaz.Etemad@uibk.ac.at (S.E.); Eva.Albertini@uibk.ac.at (E.A.); szalkabea@gmail.com (B.K.-S.); Max.Holzknecht@uibk.ac.at (M.H.); Elia.Cappuccio@uibk.ac.at (E.C.); Maria.Cavinato-Nascimento@uibk.ac.at (M.C.)

**Keywords:** FAHD1, reprogramming, iPSCs, mitochondria, glycolytic activity, oxidative phosphorylation, neuronal differentiation

## Abstract

Mitochondria play a key role in metabolic transitions involved in the reprogramming of somatic cells into induced pluripotent stem cells (iPSCs), but the underlying molecular mechanisms remain largely unexplored. To obtain new insight into the mechanisms of cellular reprogramming, we studied the role of FAH domain-containing protein 1 (FAHD1) in the reprogramming of murine embryonic fibroblasts (MEFs) into iPSCs and their subsequent differentiation into neuronal cells. MEFs from wild type (WT) and Fahd1-knock-out (KO) mice were reprogrammed into iPSCs and characterized for alterations in metabolic parameters and the expression of marker genes indicating mitochondrial biogenesis. Fahd1-KO MEFs showed a higher reprogramming efficiency accompanied by a significant increase in glycolytic activity as compared to WT. We also observed a strong increase of mitochondrial DNA copy number and expression of biogenesis marker genes in Fahd1-KO iPSCs relative to WT. Neuronal differentiation of iPSCs was accompanied by increased expression of mitochondrial biogenesis genes in both WT and Fahd1-KO neurons with higher expression in Fahd1-KO neurons. Together these observations establish a role of FAHD1 as a potential negative regulator of reprogramming and add additional insight into mechanisms by which FAHD1 modulates mitochondrial functions.

## 1. Introduction

Induced pluripotent stem cells (iPSCs) are promising cellular sources for ex vivo disease modeling, drug screening, and upcoming applications in regenerative medicine. iPSCs self-renew without limit in tissue culture and can form any cell type. Since their introduction in 2006, iPSCs have become a major focus for both basic and applied research, in part, because of their unique growth characteristics and cellular properties, and their high potential in personalized medicine without the ethical implications carried by the derivation of embryonic stem cells [1].

Somatic cell reprogramming to iPSCs includes a transition in mitochondrial morphology from an elongated, filamentous, and branching network structure to a collection of small, punctate, and separate organelles [2]. Concurrent with this morphologic shift, metabolism is switched from mainly OXPHOS in somatic cells towards mainly glycolytic metabolism in reprogrammed iPSCs. Additional changes that occur during reprogramming include a reduction in mitochondrial (mt) DNA copy number, alterations in the tricarboxylic acid (TCA) cycle metabolite levels, changes in Ca^2+^ handling, the production of Fe–S clusters, and an obligatory time-coordinated oxidative burst [2]. Many stem cells exhibit low mitochondrial content and reliance on mitochondrial-independent glycolytic metabolism for energy. However, accumulating evidence indicates that mitochondria play an important role in regulating stem cell activity, fate decisions, and defense against senescence [3].

It is known that mitochondrial dysfunction can disrupt these key metabolic transitions and may result in incomplete reprogramming, spontaneous differentiation [4], or cell death. The Jansen-Dürr lab recently identified fumarylacetoacetate hydrolase (FAH) domain-containing protein 1 (FAHD1), a member of the FAH superfamily, which unlike the founding member fumarylacetoacetate hydrolase (FAH), plays no role in tyrosine catabolism. Instead, FAHD1 was identified as a mitochondrial metabolic enzyme with dual substrate specificity, which can hydrolyze both acylpyruvates [5] and decarboxylate oxaloacetate [6], a metabolite of the TCA cycle. Notably, FAHD1 is required for sustained mitochondrial function in human cells; accordingly, depletion of FAHD1 from human endothelial cells impairs mitochondrial function which subsequently induces mitochondrial dysfunction-induced cellular senescence (MiDAS) [7] by causing metabolic alterations independent of the DNA damage response pathway [8,9]. 

Given the key roles of glycolysis and mitochondrial metabolism in reprogramming and neuronal differentiation, respectively, here we analyze the role of FAHD1 in these biological processes, using well established protocols for iPSC generation and subsequent neuronal differentiation [10,11]. To address the role of FAHD1 in this context, mouse embryonic fibroblasts (MEFs) were generated from Fahd1-knock-out (KO) and wild type (WT) mice and used for the establishment of iPSCs. We also compared their reprogramming efficiency, germ layer and neural differentiation, mt DNA content, and mt biogenesis as well as OXPHOS activity and glycolysis.

## 2. Materials and Methods

### 2.1. Generation of iPSCs

MEFs were isolated from C75BL/6N FAHD1^-/-^ and WT mice at E13.5 according to an established protocol [12]. MEFs were cultured in fibroblast medium, consisting of DMEM (high glucose, pyruvate; Gibco, Waltham, MA, USA, 41966029) supplemented with 10% fetal bovine serum (Gibco, 10270106), 1% penicillin/streptomycin (Sigma-Aldrich, Darmstadt, Germany, P4333), 2 mM L-Glutamine (Gibco, 25030081), 1% nonessential amino acids (Sigma-Aldrich, M7145), and 100 µM beta-mercaptoethanol (Gibco, 21985023). The reprogramming of fibroblasts was adapted from [13]. Retroviral vectors containing Oct3/4 (Addgene, Watertown, MA, USA, 13366), Sox2 (Addgene 13367), Klf4 (Addgene 13370), c-Myc (Addgene 13375), and eGfp (Addgene 13826) were transfected into Platinum-E Retroviral Packaging Cell Line (Cell Biolabs, San Diego, CA, USA, RV101) using the calcium phosphate transfection method [14]. After 48 h, supernatants containing viral particles (two virus stocks, 24 h transfection each) were applied to the KO and WT MEFs in the presence of 4 µg/mL polybrene (Merck, Darmstadt, Germany, TR-1003-G). Twenty-four hours after the second infection, the virus-containing medium was replaced with stem cell medium containing KnockOut™ DMEM (Gibco, 10829018), 10% KnockOut™ Serum Replacement (Gibco, 10828010), 2 mM L-Glutamine (Gibco, 25030081), 0.1 mM non-essential amino acids (Sigma-Aldrich, M7145), 0.1 mM beta-mercaptoethanol (Gibco, 21985023), and 10^3^ units/mL leukemia inhibitory factor LIF (Gibco, PHC9484). The medium was changed every day until a change in morphology and iPSC-like colony formation was observed. During this process, iPSC-like colony formation was observed using a motorized Leica DMi8 fluorescent inverted microscope equipped with a Hamamatsu Orca Flash 4.0 V2 camera. Tile scan images were taken to cover the majority of the well surface and colonies were counted from these images using ImageJ software 1.53c. Individual colonies were then manually picked under a fluorescent microscope (EVOS XL Core Imaging System) and clonally expanded on an irradiated MEF feeder layer (Gibco, A24903). Three Fahd1-KO and three WT iPSC-like clones were then selected for the subsequent analysis.

### 2.2. iPSC Differentiation

For embryoid body (EB) formation and germ layer differentiation, iPSCs were harvested with Accutase (Sigma-Aldrich, A6964) and cultured on non-adherent bacterial dishes in embryoid body medium consisting of KnockOut™ DMEM (Gibco, 10829018) with 20% KnockOut™ Serum Replacement (Gibco, 10828010), 2 mM L-Glutamine (Gibco, 25030081), and 1% non-essential amino acids (Sigma-Aldrich, M7145). After 4 days of suspension culture, the EBs formed were transferred onto 0.1% gelatine-coated 12-well plates with coverslips and cultured for another 10 days. The medium was changed every second day.

For neuronal differentiation we followed a classical five-stage protocol [10]. Briefly, WT and Fahd1-KO iPSCs were grown on 0.1% gelatine-coated tissue culture plates in the presence of 1400 U/mL of leukemia inhibitory factor (LIF; Gibco, PHC9484) in stem cell medium. After 4 days in suspension without LIF, the EBs formed were then plated and cultured for 6 days onto adhesive tissue culture surface with neural selection medium consisting of DMEM/F12 (Gibco, 21331020), 1.5 mg/mL glucose (Sigma-Aldrich, G8644), 0.24% NaHCO3 (Gibco, 25080094), 2 mM L-Glutamine (Gibco, 25030081), 5 mg/mL fibronectin (Sigma-Aldrich, F1056), and 1x insulin/transferrin/sodium selenite supplement (Sigma-Aldrich, I9278). At this point, neural progenitor cells were visible and formed neural rosettes. For neuronal differentiation, cells were dissociated with Accutase (Sigma-Aldrich, A6964) and plated on GFR Matrigel- (Corning, 354230) coated tissue culture dishes in N2 medium consisting of 50:50 DMEM/F12/Neurobasal (Gibco, 21331020 and Gibco, 21103049, respectively), 1:50 B-27 supplement (Gibco, 17504044), 2 mM L-Glutamine (Gibco, 25030081), 1% ascorbic acid (Sigma-Aldrich, A4403), and 1% N-2 supplement (Gibco, 17502048), with 3 mM purmorphamine (Miltenyi, 130-104-465) and 10 ng/mL bFgf (Gibco, 13256-029). After 4 days, Fgf2 and purmorphamine were withdrawn from the N2 medium and cells were cultured for an additional 6 days.

For mitochondrial DNA analysis and OXPHOS and glycolysis experiments, the iPSCs were adapted to feeder-free conditions in 2i medium consisting of 1:1 DMEM/F12 and Neurobasal (Gibco, 21331020 and Gibco, 21103049, respectively), 1% N-2 supplement (Gibco, 17502048), B-27 supplement (Gibco, 17504044), 18 mg/mL BSA V (Applichem, A1391), 2 mM L-Glutamine (Gibco, 25030081), 1.5 × 10^−4^ M monothioglycerol (Sigma-Aldrich, M1753), 10^6^ U/mL LIF (Gibco, PHC9484), 1 µM PD03259010 (STEMCELL Technologies, Köln, Germany, 72182), and 3 µM CHIR99021 (Axon Medchem, Groningen, The Netherlands, 1386). To do so, iPSCs were split at 1:3 ratio on gelatine-coated dishes in stem cell medium to get rid of the feeder cells. After two passages, the serum-containing medium was replaced with 2i medium and splitting of the cells was performed every 2 days at 1:3 ratio. The iPSCs were adapted to the 2i medium after four to six passages. Medium was changed every 2 days.

### 2.3. MEF Growth Curve

First, 5 × 10^5^ P0 MEF cells Fahd1-KO and WT MEFs were plated in triplicate. Cell proliferation was evaluated using a CASY^®^Cell counter & AnalyzerSystem (OMNI Life Science, Bremen, Germany) at 4, 9, 13, 19, 23, 30, and 37 days in vitro (DIV). For this purpose, MEFs were washed in PBS, detached in trypsin-ethylenedia-minetetraacetic acid (EDTA) 0.05% solution, and counted using CASY. Post counting at each given DIV, 5 × 10^5^ KO and WT MEF cells were plated again to repeat the procedure. Cumulative population doubling (PDL) was calculated as previously described [15].

### 2.4. qRT-PCR

Cultured cells were homogenized in 1 mL of TRI Reagent (MRC, TR 118). RNA was isolated following the manufacturer’s instructions. The precipitated RNA pellet was dissolved in 20 µL of nuclease-free water and reverse-transcription was performed using Maxima cDNA Synthesis Kit (Thermo Scientific, Vienna, Austria, K1672) according to the manufacturer’s instructions. Gene expression was assessed by qRT-PCR on a Bio-Rad CFX Connect Real time system using SYBR Green (Solis Biodyne, Tartu, Estonia, 08-24-00008) as the detection method, and the geometric mean of the reference gene Gapdh as internal normalization reference. 

For mtDNA/nDNA ratio, genomic DNA extraction and ratio calculation was performed according to [16]. A concentration of 10 ng/µL DNA was used for qRT-PCR. Different genes were selected to evaluate the relative copy number of mt DNA and nuclear (n) DNA. In the mouse mitochondrial genome, genes corresponding to the stable fraction that is not prone to deletions encode for 16S rRNA and ND1 [17]. Hexokinase 2 (Hk2) and Gapdh are nuclear encoded genes that we selected for our assay. A comparison of Nd1 and 16S rRNA DNA expression relative to Hk2 or Gapdh DNA expression will give a measure of mt DNA copy number to n DNA copy number ratio. Analysis was calculated by following the classical ΔΔCt method.

The forward and reverse primer sequences used: Gapdh (FWD: 5′-AGGGCTCATGACCACAGTC-3′; REV: 5′-CAGCTCTGGGATGACCTTG-3′), Nanog (FWD: 5′-CCACCAGGTGAAATATGAGAC-3′; REV 5′-TATTTGGAAGAAGGAAGGAACC-3′), Pax6 (FWD: 5′-ACACCTGTCTCCTCCTTCAC-3′; REV: 5′-GGTTGCATAGGCAGGTTGTTTG-3′), Nestin (FWD: 5′-AAGTGGCTACATACAGGACTC-3′; REV: 5′-TGAGGACAGGGAGCACAGA-3′), 16S rRNA (FWD: 5′-CCGCAAGGGAAAGATGAAAGAC-3′; REV: 5′-TCGTTTGGTTTCGGGGTTTC-3′), ND1 (FWD: 5′-CTAGCAGAAACAAACCGGGC-3′; REV: 5′-CCGGCTGCGTATTCTACGTT-3′), HK2 (FWD: 5′-GCCAGCCTCTCCTGATTTTAGTGT-3′; REV: 5′-GGGAACACAAAAGACCTCTTCTGG-3′).

### 2.5. Immunocytochemistry

Cells plated on glass coverslips were washed in 0.1 M PBS and fixed in 4% paraformaldehyde (Sigma-Aldrich, 151827) dissolved in 0.1 M PBS. After washing, the cells were permeabilized with 0.3% Triton X-100 (Sigma-Aldrich, X100) in 0.1 M PBS for 15 min and then incubated for 1 h in the blocking solution containing 0.025% Triton X-100 and 10% fetal bovine serum (Gibco, 10270106) in 0.1 M PBS. Subsequently, cells were incubated overnight at 4 °C with the primary antibodies diluted in the blocking solution containing 0.025% Triton X-100 and 5% fetal bovine serum in 0.1 M PBS. On the next day, cells were washed in 0.1 M PBS/0.025% Triton X-100 (PBST) and incubated with the corresponding fluorescent secondary antibodies (Thermo Scientific). After washing with PBST, the cell nuclei were stained with DAPI (Invitrogen, Waltham, MA, USA, D21490) for 5 min and then washed again with distilled water. Finally, coverslips were mounted onto adhesive slides using Aqua-Poly/Mount (Polysciences, Warrington, PA, USA, 18606). The primary antibodies used were: mouse monoclonal anti-alpha smooth muscle actin SMA (Abcam, Cambridge, UK, ab7817); mouse monoclonal anti-tubulin beta 3 or Tuj1 (Biolegend, San Diego, CA, USA, 801202); rabbit polyclonal alpha-1-Fetpoprotein AFP (Agilent, Santa Clara, CA, USA, A000829-2); rabbit polyclonal anti-Oct4 (GeneTex, Alton Pkwy Irvine, CA, USA, GTX101497); and mouse monoclonal Sox2 (Bio-techne, Minneapolis, MN, USA, MAB2018). The stained cells were observed using a motorized Leica DMi8 fluorescent inverted microscope equipped with a Hamamatsu Orca Flash 4.0 V2 camera.

### 2.6. Mitochondrial and Glycolysis Stress Tests

Real-time monitoring of oxygen consumption rate (OCR) and extracellular acidification rate (ECAR) was performed using a Seahorse XFp Analyzer, applying the Seahorse XFp Cell Mito Stress Test Kit and the Seahorse XFp Glycolysis Stress Test Kit (Agilent, Santa Clara, CA, USA) according to the manufacturer’s protocols. Briefly, 5 × 10^4^ MEFs or feeder-free iPSCs were seeded onto XFp cell culture miniplates the day before the experiment. The medium was changed to a special assay medium according to the manufacturer’s protocol 60 min before start of the experiment. For the mitochondrial stress test, cells were incubated in the mito stress test assay medium. OCR and ECAR were reported under basal conditions and in response to ATP synthase inhibitor oligomycin (2 µM), mitochondrial uncoupler FCCP (0.5 µM), and the complex I and III inhibitors rotenone and antimycin A (0.5 µM). For the glycolysis stress test, cells were incubated in the glycolysis stress test assay medium without glucose or pyruvate prior to measuring. ECAR was reported under basal conditions and in response to glucose (10 mM), ATP synthase inhibitor oligomycin (2 µM), and glycolysis inhibitor 2-deoxy-glucose (2-DG, 50 mM). OCR is reported as pmol/min and ECAR as mpH/min. Values obtained for both ECAR and OCR were normalized to cell number in all cases. Using the Seahorse Report Generator; mitochondrial and glycolysis stress tests assay parameter values such as glycolysis (maximum rate measurement before oligomycin injection)—(last rate measurement before glucose injection), glycolytic capacity (maximum rate measurement after oligomycin injection)—(last rate measurement before glucose injection), and glycolytic reserve (glycolytic capacity)—(glycolysis) and other parameters were automatically calculated, while non-glycolytic acidification (last rate measurement prior to glucose injection) was automatically measured. 

### 2.7. Statistical Analysis

For statistical analysis, two-tailed unpaired Student’s t test was used for the comparison of the mean values. In the figures, * *p* < 0.05, ** *p* < 0.01, and *** *p* < 0.001. Error bars represent the standard error of the mean (SEM) in all bar graphs. Three to six independent biological replicates were included in all analyses. 

## 3. Results

### 3.1. Reprogramming of WT and Fahd1-KO MEFs

To assess the impact of Fahd1 on stem cell reprogramming, we reprogrammed MEFs derived from WT and Fahd1-KO mice using retroviral infection with the four factors Oct4, Sox2, c-Myc, and Klf4. We also included eGFP for controlling the reprogramming efficiency and observing the colony formation. The reprogramming experiments were conducted on three independent WT and three independent Fahd1-KO MEF lines, derived from three different WT and Fahd1-KO mouse embryos, respectively. At D1 post infection, the percentage of infected MEFs, as indicated by the GFP fluorescence, was similar between WT and Fahd1-KO MEFs (Appendix A). Starting from day 3 (D3) post infection, we observed a dramatic increase in the formation of colonies in Fahd1-KO-infected MEFs as compared to WT based on the number of GFP-positive colonies (Figure 1A,B). We also counted all colonies formed where cells were compacted with a dome shaped morphology (Appendix A), both fluorescent and non-fluorescent, at days 3, 4, 5, and 7. The results showed a significant 50% increase in the number of colonies in Fahd1-KO MEFs in comparison to WT (Figure 1C). This increase was not due to a difference in cell proliferation between MEF lines derived from WT versus KO mice, since no difference was observed in the growth curves of KO and WT MEFs (Figure 1D). Three iPSC-like clones from each of the six MEF lines (three Fahd1-KO and three WT) were selected for the subsequent analysis.

### 3.2. Germ Layer and Neural Differentiation

We next analyzed the pluripotency status of the selected clones. Both KO and WT iPSCs showed typical round stem cell colonies when cultured in vitro on the feeder layer (Figure 2A). Immunocytochemistry showed that they both co-expressed the pluripotency markers Oct4 and Sox2 (Figure 2B). Germ layer differentiation also confirmed the ability of KO and WT iPSCs to differentiate into the three germ layers, as indicated by the expression of the endodermal marker AFP, the mesodermal marker SMA, and the ectodermal marker Tuj1 (Figure 2C). The most common pluripotency markers Nanog, Oct4, and Sox2 were similarly expressed in iPSCs of either genotype (Figure 2B,E). To compare cell fate decisions, we differentiated the iPSCs toward a neural fate [10]. KO-Fahd1 and WT iPSCs formed in vitro the typical neural rosettes morphology during neural specification, and they both displayed an increased expression of the neural progenitor markers Pax6 and Nestin with a downregulation of the pluripotency marker Oct4 (Figure 2D,E). As expected, expression of Fahd1 was not detectable in any of the Fahd1-KO cell types, and this was also confirmed by Western blot analysis (Appendix A). Together, these findings suggest that Fahd1 deficiency does not grossly affect the differentiation of iPSCs to neural progenitor cells.

### 3.3. Glycolytic Activity Is Enhanced in Fahd1-Deficient Mouse Embryonic Fibroblasts

In order to decipher the cause of the increased iPSC-like colony formation in Fahd1-KO MEFs during reprogramming (Figure 1), we first measured glycolytic activity in Fahd1-KO and WT MEFs, using a Seahorse Extracellular Flux analyzer. To this end, we analyzed the extracellular acidification rate (ECAR), using a glycolysis stress assay. The assay measures changes in extracellular acidification rate upon serial injection of glucose, ATP-synthase inhibitor oligomycin, and glycolysis inhibitor 2-DG and thus allows assessment of the key parameters of glycolytic flux: glycolysis, glycolytic capacity, and glycolytic reserve as well as non-glycolytic acidification. Interestingly, glycolysis as well as non-glycolytic acidification, glycolytic capacity, and glycolytic reserve were all significantly higher in Fahd1-KO MEFs as compared to WT. In particular, extracellular acidification due to glycolytic activity increased from 17.5 ± 1.3 mpH/min/cell in WT MEFs to 25.6 ± 1.2 mpH/min/cell in Fahd1-KO MEFs, accompanied by a similar increase in glycolytic capacity (Figure 3A). Next, we assessed mitochondrial function by the Seahorse Extracellular Flux analyzer using a MitoStress test kit. This kit uses modulators of cellular respiration that specifically target components of the electron transport chain (ETC) to reveal key parameters of metabolic function. The compounds, oligomycin, FCCP, and a mix of rotenone and antimycin A, are serially injected to measure ATP-linked respiration, maximal respiration, and non-mitochondrial respiration, respectively. Proton leak and spare respiratory capacity are then calculated using basal respiration and these parameters. Whereas routine respiration, measured through oxygen consumption rate (OCR), was not significantly different between WT MEFs (51.1 ± 2.8 pmol O_2_/min/cell) and Fahd1-KO MEFs (47.7 ± 2.2 pmol O_2_/min/cell), we observed a trend for reduced mitochondrial respiration in Fahd1-KO relative to WT MEFs, which reached statistical significance for two parameters, namely maximal respiration and spare respiratory capacity. In particular, maximal respiration was decreased from 131.6 ± 7.5 pmol O_2_/min/cell in WT MEFs to 102.6 ± 9.1 pmol O_2_/min/cell in Fahd1-KO MEFs (Figure 3B), suggesting a decreased respiratory capacity of Fahd1-KO MEFs.

To assess if there are any changes in mitochondrial copy number in Fahd1-KO MEFs as compared to WT, we determined the relative amount of mitochondrial versus nuclear DNA (mtDNA/nDNA ratio). When the expression of two mt genes 16S and ND1 was normalized to two nuclear housekeeping genes (Hk2, Gapdh), we found no discernable difference in the relative mtDNA/nDNA expression between WT and KO MEFs (Figure 3C).

Taken together, these observations suggest a significant increase in glycolytic metabolism in Fahd1-deficient MEFs, combined with a trend for mild mitochondrial dysfunction, reflected by decreased respiratory capacity.

### 3.4. Mitochondrial Remodeling during Reprogramming Is Affected by Fahd1 Deficiency

We also determined glycolytic activity in iPSCs derived from Fahd1-KO and WT MEFs, using the glycolysis stress assay. It is well established that reprogramming of somatic cells to iPSCs involves a significant increase of glycolytic activity [2]. As expected, reprogramming of WT MEFs increased ECAR due to glycolytic activity from 17.5 ± 1.3 mpH/min/cell (Figure 3A) to 45.1 ± 2.1 mpH/min/cell in the resulting iPSCs (Figure 4A); similarly, reprogramming of Fahd1-KO MEFs increased ECAR due to glycolytic activity from 25.6 ± 1.2 mpH/min/cell (Figure 3A) to 53.5 ± 2.3 mpH/min/cell in the resulting iPSCs (Figure 4A), which was still significantly higher compared to the iPSCs derived from WT MEFs. 

It has been described that during reprogramming, mitochondrial function is decreased, including reduced OXPHOS activity, a reduction in the mt DNA copy number, and a more punctate mitochondrial morphology [2]; however, others have described that mitochondrial function is still important in iPSCs (see Section 4). When mitochondrial function was analyzed in iPSC clones using MitoStress Kit, we observed a significant increase of basal respiration from 47.7 ± 2.2 pmol O_2_/min/cell (Figure 3B) to 163.2 ± 17.1 pmol O_2_/min for WT iPSCs (Figure 4B); in the case of Fahd1-KO MEFs, reprogramming led to an even higher basal respiration, amounting to 307.6 ± 8.4 pmol O_2_/min/cell. These changes were accompanied by a similar increase of the maximal respiratory capacity in Fahd1-KO vs. WT iPSCs (Figure 4B). When mt DNA copy number was assessed in iPSCs, we observed a significant increase in the mtDNA/nDNA ratio in Fahd1-deficient relative to WT cells, suggesting that a mitochondrial imbalance occurs during reprogramming of Fahd1-deficient vs. WT cells (Figure 4C). This could be due to increased mitochondrial biogenesis in Fahd1-deficient cells, an increased mitochondrial turnover in WT cells, or a combination of both effects. A general reduction of mt DNA copy number has been described to occur during reprogramming in several cell types. In concordance with these studies, we also observed a reduction in mtDNA/nDNA ratio in WT iPSCs as compared to WT MEFs, assessed by a downregulation of ND1 expression (Appendix A). Interestingly, in Fahd1-KO iPSCs a significant increase of the mt genes 16S rRNA and ND1 was observed as compared to Fahd1-KO MEFs, showing therefore an increase in mt DNA copy number in these cells, at least partially (Appendix A). It is conceivable that the mt DNA reduction in WT may be counteracted by increased mitochondrial biogenesis in the case of Fahd1-deficient iPSCs. In support of this hypothesis, we found that the expression of Pparg and Ppara, genes involved in mitochondrial biogenesis [18], was significantly upregulated in Fahd1-deficient relative to WT iPSCs (Figure 4D).

When OCR was normalized to the mt DNA copy number, basal respiration did not significantly differ between iPSCs of both genotypes; however, we observed a significant decrease in respiratory capacity in iPSCs derived from Fahd1-KO MEFs vs. WT MEFs; importantly, these experiments also revealed a significant decrease in mitochondrial coupling efficiency in iPSCs derived from Fahd1-KO MEFs vs. WT MEFs (Figure 4E). Together these data document a major difference in mitochondrial remodeling during reprogramming in Fahd1-deficient vs. WT cells, characterized by a partially impaired mitochondrial function in Fahd1-KO iPSC, compensated for by increased mt DNA copy number.

### 3.5. Higher Mitochondrial Biogenesis Is Maintained in iPSC-Derived Fahd1-KO Neurons Relative to WT

Whereas reprogramming of somatic cells to iPSCs demands the transition from mitochondrial oxidative phosphorylation to glycolysis, differentiation of iPSCs involves additional remodeling of the mitochondrial network for increased mitochondrial activity [19]. It is known that the transition from neural stem cells (NSC) to a neuronal lineage is accompanied by increased mitochondrial biogenesis, as well as the downregulation of glycolysis and fatty acid oxidation pathways [20]. For instance, the progression from pluripotent progenitor cells to neurons is characterized by a strong reduction in glycolysis-related proteins, such as hexokinase 2 (HK2) and isoform A of lactate dehydrogenase (LDHA), which metabolizes the reduction of pyruvate to lactate. Additionally, a switch from PKM2 to its constitutively active isoform PKM1 and an upregulation of OXPHOS-related genes has been observed [21,22]. When we differentiated WT and Fahd1-KO iPSCs to neurons, we observed an efficient neuronal differentiation in both genotypes as indicated by the expression of the neuronal marker Tuj1 (Figure 5A). We analyzed the mRNA expression of Fahd1 in MEFs, iPSCs, and neurons only in WT, since the mRNA expression of Fahd1 in KO was not detected. As compared to MEFs, we observed a downregulation of Fahd1 mRNA expression in iPSCs and an increase in differentiation to neurons (Appendix A). This dynamic Fahd1 expression indicated its important role in the processes we investigated. When we analyzed the relative mRNA expression of six genes (Pparg, Ppara, Slit1, Creb1, Nrf1, and Junk1) which are known to be involved in mitochondrial biogenesis [18], we found consistent upregulation of five of these genes during neuronal differentiation, irrespective of the genotype (Figure 5B,C). Nevertheless, mRNA levels of several of these genes, including Pparg, Ppara, Slit1, Nrf1, and JNK1 were higher in Fahd1-deficient neurons relative to WT, suggesting that increased mitochondrial biogenesis continues in these cells after differentiation from iPSCs to neurons (Figure 5D). These results support the conclusion that the higher degree of mitochondrial biogenesis in Fahd1-KO relative to WT cells is maintained throughout neuronal differentiation, probably compensating for mild mitochondrial dysfunction in neurons derived from Fahd1-deficient cells.

## 4. Discussion

FAHD1 is a mitochondrial metabolic enzyme playing an important role in the TCA cycle and is required for sustained mitochondrial function in human endothelial cells. On the other hand, the role of FAHD1 in stem cell reprogramming and differentiation is unknown. Here we investigated this role by reprogramming WT and Fahd1-KO MEFs into iPSCs and comparing their reprogramming efficiency. We also differentiated the derived WT and Fahd1-KO iPSCs into neurons to study the potential effects of Fahd1-KO on stem cell fate.

### 4.1. Increased Reprogramming Efficiency of Fahd1-KO MEFs Correlates with Higher Glycolytic Activity

When WT and Fahd1-KO MEFs were reprogrammed into iPSCs [13], we observed an enhanced reprogramming efficiency in Fahd1-KO MEFs demonstrated by a significant increase in the number of iPSC-like colonies as compared to WT, already from day 3 post-infection, and this difference remained visible throughout the reprogramming process. The observed increase was not due to an accelerated growth of MEFs, and also was independent of the iPSC growth rate since we were counting the number of colonies formed and not the cell number. It has been previously shown that enhancing the metabolic shift toward glycolysis facilitates reprogramming into both human and mouse iPSCs, whereas blockade of the metabolic shift prevents reprogramming [23]. When glycolysis was inhibited by 2-deoxy-D-glucose or oxalate in somatic cells, reprogramming into iPSCs was also inhibited. In contrast, stimulating glycolysis with compounds such as glucose, D-fructose-6-phosphate, 2, 4-dinitrophenol, and N-oxaloylglycine enhanced reprogramming [24,25,26]. In the current study, we demonstrated that glycolysis is highly increased in Fahd1-KO MEFs which may account for the enhanced iPSC reprogramming in these cells, in concordance with the aforementioned studies. Whereas no significant reduction of basal respiration was observed in Fahd1-KO MEFs relative to WT, we noted a significant reduction in maximal respiratory capacity in Fahd1-KO MEFs (Figure 3B), pinpointing initial mitochondrial dysfunction at this stage. It is well established in the field that inhibition of mitochondrial oxidative phosphorylation, by various different manipulations, leads to an increase in glycolysis in most cells, in order to compensate for reduced mitochondrial ATP production [27].

### 4.2. Fahd1 Deficiency Affects Mitochondrial Remodeling during Reprogramming to iPSCs

Metabolic alterations in stem cells, including iPSCs, have been described in the literature; however, so far, no consistent pattern has been established for metabolic rearrangements in naturally occurring embryonic stem cells and iPSCs. Embryonic stem cells are characterized by increased glycolytic activity and low OXPHOS activity [2], furthermore most reported studies describe stem cells as glycolytic [3,28]. In fact, it has been demonstrated that both human and murine embryonic stem cells are highly dependent on glycolytic metabolism for stem cell maintenance and/or self-renewal [29,30]. In addition, intracellular ATP levels in pluripotent stem cells (PSCs) are more affected by inhibition of glycolysis than by inhibition of mitochondrial oxidative phosphorylation (OXPHOS) [23,31]. When OXPHOS is inhibited with oligomycin in PSCs, intracellular ATP decreases only marginally, indicating a minimal role of mitochondrial OXPHOS in these cells [31]. However, ATP levels decrease in mouse embryonic stem cells after inhibition of complex III by antimycin A [29,32]. 

Similar to embryonic stem cells, iPSCs [24,33,34] are highly dependent on glycolytic metabolism for stem cell maintenance and/or self-renewal. However, reprogramming of somatic cells to iPSCs involves a more complex metabolic remodeling. Several studies have shown that iPSCs, at least partly rely on OXPHOS as a source of energy. It was documented that human and mouse fibroblasts reprogramming into iPSCs requires a transient increase of OXPHOS [35,36]. When mitochondrial uncoupling is induced in PSCs, including iPSCs, using carbonyl cyanide m-chlorophenylhydrazone (CCCP), a reduction in intracellular ATP levels and proliferation rates has been reported [29,31,32]. In this respect, it is interesting to note that during the progression from MEFs to iPSCs the difference in ECAR between WT vs. Fahd1-KO cells became less pronounced, possibly due to the fact that in iPSCs OXPHOS contributes significantly to ATP production. In support of this conclusion, we observed an increased OCR in iPSCs of both genotypes (Figure 4).

Unlike in human endothelial cells, where acute lentiviral knockdown of FAHD1 results in pronounced mitochondrial dysfunction [8], we observed only minor deficiencies in mitochondrial function in Fahd1-KO relative to WT MEFs. These findings suggest that effects of Fahd1-KO on mitochondrial function are not very pronounced in MEFs compared to other cell types (including MEF-derived iPSCs; see below). The reasons for this different outcome of Fahd1 gene silencing in different cell types require further investigation. Of note, upon reprogramming we observed a significant increase in OXPHOS, probably reflecting the dependency of iPSCs on mitochondrial ATP production. Under these conditions, we noted both an increase in mt DNA copy number in Fahd1-KO cells and a strong increase in oxygen consumption. These observations suggest that in iPSCs mitochondrial function is essential, and, in these circumstances, the lack of Fahd1 has a significant impact on mitochondrial quality in iPSCs, supported by our observation that both respiratory capacity and mitochondrial coupling rate are decreased in Fahd1-KO iPSCs. Overall, these observations point to a Fahd1-dependent mitochondrial function in iPSCs, which is compensated in Fahd1-deficient iPSCs by an increase in mitochondrial biogenesis to increase the mitochondrial biomass. Increased mt DNA copy number, as observed in Fahd1-deficient iPSCs, may also reflect impaired mitophagy, which is currently unclear; more work will be required to clarify this point.

### 4.3. Fahd1 Deficiency Does Not Grossly Affect Pluripotency of iPSCs after Reprogramming nor Their Differentiation to Neurons

Both WT and Fahd1-KO iPSCs showed similar iPSC morphology and high expression of the pluripotency markers Oct4 and Sox2. Also, iPSCs of both genotypes similarly expressed mesodermal, endodermal, and ectodermal markers when germ layer differentiation was performed. Together, these results indicate that Fahd1 deficiency does not affect the pluripotency of iPSCs. 

Various studies have provided evidence that the differentiation of PSCs is associated with a metabolic shift, where a transition from a predominant glycolysis-based metabolism in PSCs toward an increased OXPHOS-based metabolism in differentiated cells is observed [23,37,38,39]. In particular, it has been shown that neural stem cells (NSCs) rely preferentially on OXPHOS as a source of energy [39,40]. Based on these studies, we decided to differentiate in vitro WT and Fahd1-KO iPSCs toward neurons [10,41] in order to investigate if Fahd1 deficiency would impair neuronal fate in this experimental model. Interestingly, neural differentiation of iPSCs was apparently independent of Fahd1, as both WT and Fahd1-KO showed similar NSC morphology and comparable expression of the neural progenitor markers Pax6 and Nestin. On the other hand, previous studies showed an induction of mitochondrial biogenesis during differentiation of PSCs toward various cell lineages [42,43,44]. When analyzing the mRNA expression of the biogenesis markers in mature neurons, we observed a significant increase not only of Pparg but also of Slit1, Nrf1, and Junk1, indicating a further enhancement of mt biogenesis in Fahd1-KO iPSC-derived neurons. It should be noted that neuronal differentiation in cell culture conditions differs significantly from the in vivo environment, in particular concerning the availability of fatty acids and other carbon sources, which may affect neuronal differentiation in Fahd1-deficient mammals. Future in vivo studies will be essential to clarify this point.

Even though an increase of mt biogenesis is to be expected during differentiation, the expression of the main regulators of mitochondrial biogenesis [43] was significantly higher in Fahd1-KO neurons compared to WT. These findings indicate that mitochondrial biogenesis is increased in Fahd1-KO neurons, in order to compensate for the disrupted mitochondrial function in these cells, similar to our findings with Fahd1-KO iPSCs. The precise role of FAHD1 in neuronal differentiation will be further addressed in future studies.

To conclude, according to our hypothesis and supported by published and unpublished experimental data, the current view is that Fahd1-KO leads to a decreased activity of complex II of the mitochondrial electron transport chain, resulting from accumulation of oxaloacetate (OAA) which is a competitive inhibitor of succinate dehydrogenase (SDH) [9] (Figure 6). This results in the accumulation of succinate, reduced levels of fumarate, and reduced flux through the TCA cycle, together leading to an impaired mitochondrial function, the extent of which differs between cell types. Severe mitochondrial dysfunction due to FAHD1 deficiency will be compensated for by increased mtDNA copy number (Figure 6).

## Figures and Tables

**Figure 1 cells-10-02040-f001:**
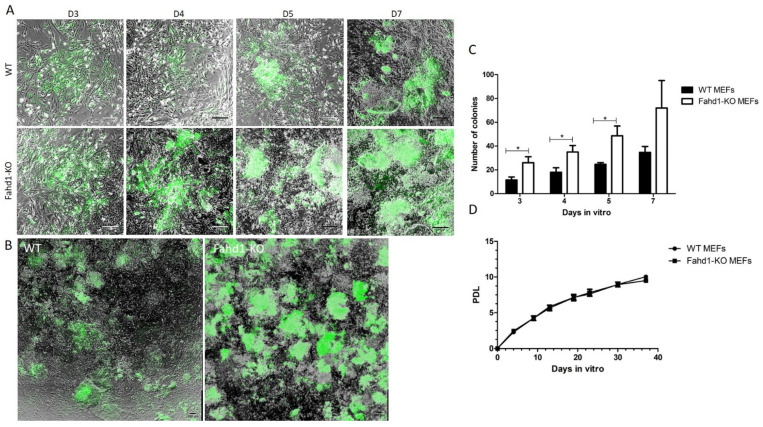
Increased iPSC-like colony formation in Fahd1-KO MEFs during reprogramming. (**A**) Merged phase-contrast and fluorescent images showing the difference in the fluorescent colony formation during the course of reprogramming from day 3 (D3) to D7 of WT and Fahd1-KO MEFs. (**B**) Merged tile scan images of phase contrast and fluorescent images covering most of the surface of a well of WT and Fahd1-KO reprogrammed MEFs, respectively, at D7. (**C**) Counting the fluorescent and non-fluorescent iPSC-like colonies at 3, 4, 5, and 7 days in vitro shows a clear and significant increase in the number of colonies formatted in Fahd1-KO as compared to WT, * *p* < 0.05, *n* = 3. (**D**) Growth curve comparison between WT and Fahd1-KO MEFs for 37 days in vitro. No significant difference was observed. MEFs: mouse embryonic fibroblasts, D: day in vitro, WT: wild type, KO: knock-out, PDL: cumulative population doubling level. Scale bars, 200 µm.

**Figure 2 cells-10-02040-f002:**
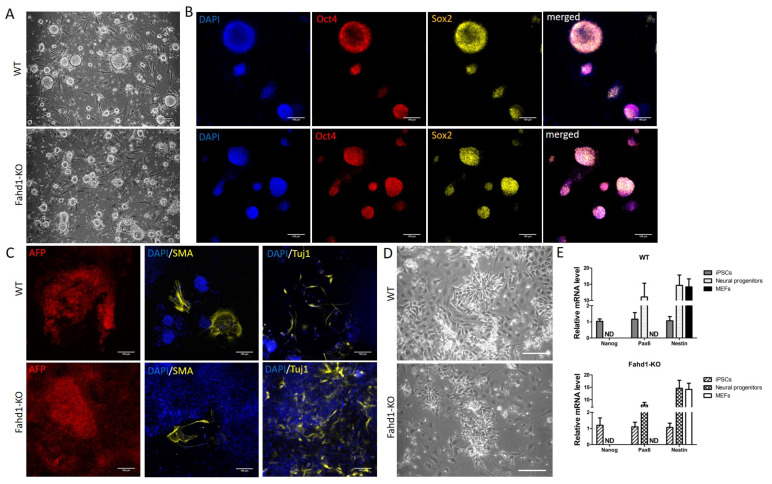
Germ layer and neural differentiation showed similar cell fate between WT and Fahd1-KO iPSCs. (**A**) Phase contrast imaging showing typical iPSC-like round colonies when cultured on inactivated MEFs. (**B**) Immunofluorescence showing the expression of the pluripotency markers Oct4 and Sox2 in both WT and KO iPSCs. Cell nuclei were stained with DAPI. (**C**) After germ layer differentiation, immunofluorescence on the differentiated cells showed the expression of the endodermal marker AFP, the mesodermal marker SMA, and the ectodermal marker Tuj1. (**D**) Neural differentiation showed typical neural rosette formation both in WT and Fahd1-KO, as it is shown in the phase contrast images. (**E**) Relative mRNA expression by qRT-PCR of the pluripotency marker Nanog, and the neural progenitor markers Pax6 and Nestin in WT and Fahd1-KO MEFs, iPSCs, and neural progenitors. The results showed a downregulation of the pluripotency and an upregulation of neural markers similarly in both WT and Fahd1-KO. MEFs: mouse embryonic fibroblasts, iPSCs: induced pluripotent stem cells, WT: wild type, KO: knock-out, ND: not detected. Scale bars, 100 µm.

**Figure 3 cells-10-02040-f003:**
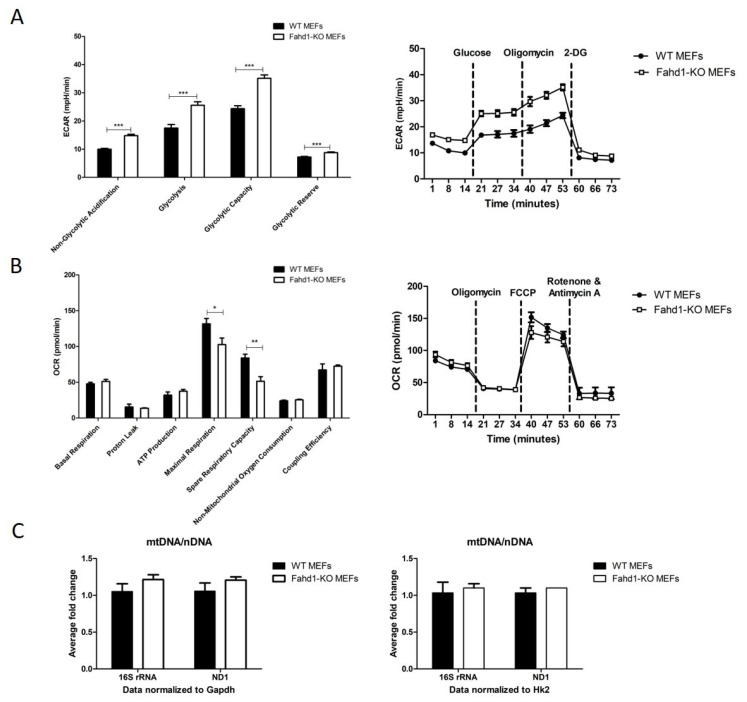
Increased glycolysis in Fahd1-KO MEFs. (**A**) ECAR measurements in mpH/min/cell using Seahorse XFp analyzer and following glycolysis stress test protocol showed a highly significant increase in basal glycolysis, glycolytic capacity, non-glycolytic acidification, and glycolytic reserve in Fahd1-KO MEFs as compared to WT, *** *p* < 0.001, n = 6. (**B**) OCR measurements in pmol/min/cell using Seahorse XFp analyzer and following MitoStress test protocol showed a trend to decreased OCR in Fahd1-KO vs. WT MEFs, whereas maximal respiration and spare respiratory capacity parameters were significantly decreased in Fahd1-KO, * *p* < 0.05, ** *p* < 0.01, n = 3. (**C**) Average fold-change of mtDNA/nDNA ratio showing the expression of the mt genes 16S rRNA and ND1 as normalized to two housekeeping genes Gapdh and Hk2. The DNA copy number of these genes was determined in WT and Fahd1-KO MEFs. No significant difference was observed, n = 6. MEFs: mouse embryonic fibroblasts, iPSCs: induced pluripotent stem cells, WT: wild type, KO: knock-out, ECAR: extra cellular acidification rate, OCR: oxygen consumption rate, mt: mitochondrial, n: nuclear.

**Figure 4 cells-10-02040-f004:**
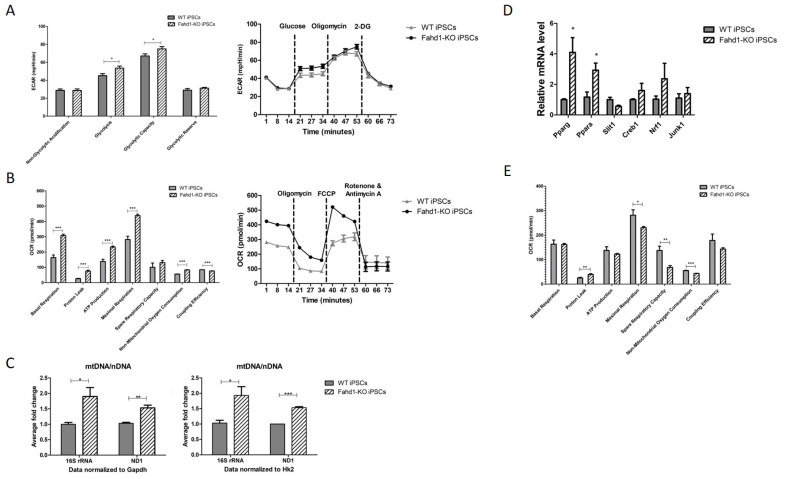
Mitochondrial remodeling during reprogramming is affected by Fahd1 deficiency. (**A**) ECAR measurements in mpH/min using the Seahorse XFp analyzer and following glycolysis stress test protocol showed that basal glycolysis, as well as glycolytic capacity, were weakly but significantly increased in Fahd1-KO iPSCs as compared to WT, * *p* < 0.05, n = 3. (**B**) OCR measurements in pmol/min using the Seahorse XFp analyzer and following MitoStress test protocol showed a significant increase of OCR for almost all parameters in Fahd1-KO iPSCs, *** *p* < 0.001, n = 3. (**C**) Average fold-change of mtDNA/nDNA ratio showing the relative concentration of mitochondrial vs. nuclear DNA, determined by qPCR amplification using primers for the mt genes coding for 16S rRNA and ND1, respectively, which were separately normalized to nuclear DNA amplified using primers for nuclear genes Gapdh and Hk2. qPCR analysis was performed in WT and Fahd1-KO iPSCs, as indicated, * *p* < 0.05, ** *p* < 0.01, *** *p* < 0.001, n = 3. (**D**) Relative mRNA expression of markers known to be involved in mt biogenesis, Pparg, Ppara, Slit1, Creb1, Nrf1, and Junk1. Their mRNA expression was evaluated in WT and Fahd1-KO iPSCs. Pparg and Ppara mRNA expression was significantly increased in Fahd1-KO iPSCs compared to WT, * *p* < 0.05, n = 3. (**E**) OCR normalized to mitochondrial copy number was slightly higher in WT vs. Fahd1-KO iPSCs, * *p* < 0.05, ** *p* < 0.01, *** *p* < 0.001, n = 3. iPSCs: induced pluripotent stem cells, WT: wild type, KO: knock-out, mt: mitochondrial, n: nuclear, OCR: oxygen consumption rate.

**Figure 5 cells-10-02040-f005:**
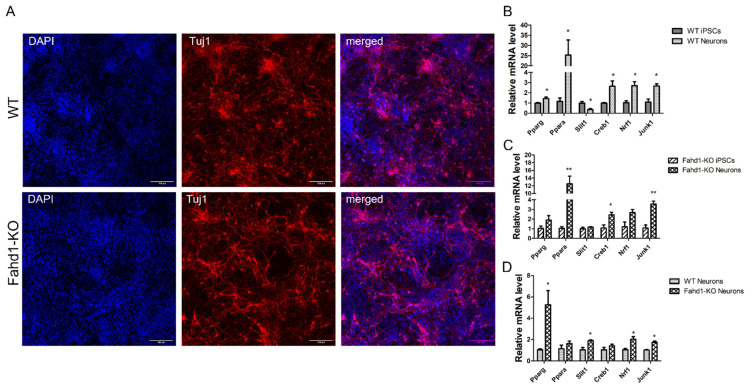
A high degree of mitochondrial biogenesis is maintained in Fahd1-KO iPSC-derived neurons. (**A**) Immunocytochemistry against the neuronal marker Tuj1 and the nuclei marker DAPI showed an efficient neuronal differentiation in both WT and Fahd1-KO iPSCs. (**B**–**D**) Relative mRNA expression of mt biogenesis markers, Pparg, Ppara, Slit1, Creb1, Nrf1, and Junk1. (**B**) Upregulation of mt biogenesis markers in iPSC-derived neurons relative to WT iPSCs. (**C**) Upregulation of mt biogenesis markers in Fahd1-KO iPSC-derived neurons relative to Fahd1-KO iPSCs. (**D**) Expression of mt biogenesis markers in neurons produced from Fahd1-KO and WT iPSCs. All the analyses were performed on iPSC-derived neurons after 20 days of differentiation, *p* < 0.05, ** *p* < 0.01, n = 3. iPSCs: induced pluripotent stem cells, WT: wild type, KO: knock-out.

**Figure 6 cells-10-02040-f006:**
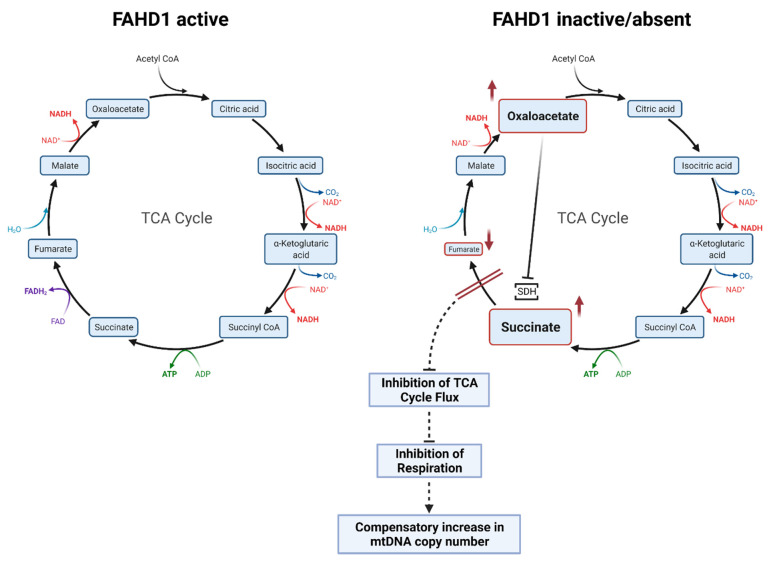
Role of FAHD1-mediated oxaloacetate decarboxylation on TCA cycle flux. When FAHD1 is functional (left panel), excess levels of oxaloacetate (OAA) are converted to pyruvate and CO_2_, keeping OAA concentration below the threshold required for efficient inhibition of succinate dehydrogenase (SDH), thus enabling sustained flux through the TCA cycle. When the FAHD1 gene is deleted, OAA may accumulate to an extent sufficient to inhibit SDH activity, resulting in the accumulation of succinate, reduced levels of fumarate, and reduced flux through the TCA cycle, together leading to an impaired mitochondrial function, the extent of which differs between cell types (see also main text) (Figure adapted from Etemad et al., Mechanisms of Ageing and Development 177 (2019) 22–29). Severe mitochondrial dysfunction due to FAHD1 deficiency will be compensated for by increased mtDNA copy number (as shown here for FAHD1-deficient iPSC; see also main text). SDH: succinate dehydrogenase. This figure as well as the graphical abstract were created with BioRender.com (accessed on 5 July 2021).

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
