# Peer review of "High Glycolytic Activity Enhances Stem Cell Reprogramming of Fahd1-KO Mouse Embryonic Fibroblasts"

_cells, 2021, doi:10.3390/cells10082040_

Round 1

Reviewer 1 Report

In this manuscript, Ahmad Salti et al described an enhanced glycolytic activity mediated elevated iPSCs induction from somatic MEFs by FAHD1 functional loss. It’s a well-designed study with extensive discussion through the glycolytic and OXPHOS topic in terms of their function in the OSKM induction of iPSCs, as well as the attempt to reach a mechanistic explanation underlying this phenotype. However, there are a few concerns that may still draw the author’s attention as listed below.

  1. In the statement of elevated induction efficiency of iPSCs from FADH1-KO MEFs, it’s impressive that the authors ruled out the possibility of accelerated growth of MEFs brought by FADH1 knockout. However, as far as I can tell, it’s very likely that the observed “elevated iPSC induction rate” can be also an artifact due to the enhanced growth rate of FADH1-KO iPSCs.
  2. Even though it’s explicitly stated in the manuscript that “Fahd1 deficiency does not affect pluripotency of iPSCs after reprogramming nor their differentiation to neurons”. However, this gene is actually known for hereditary tyrosinemia and its loss would lead to severe liver and renal disease. Thus the translational potential of FADH1-KO or inhibition in iPSCs is limited unless a transient inhibition is feasible. Additional study or discussion of tyrosine accumulation in FADH1-KO iPSC and neuron differentiation should be added.
  3. FADH1 encodes a mitochondrial enzyme with dual substrate specificity that both hydrolyze acylpyruvate and decarboxylate oxaloacetate. However, the discussion section seems never referring its function or connecting its function to the mitochondrial dysfunction, rather put them all in the a box as “mitochondrial dysfunction”. In depth discussion in this manner is required.

Author Response

In this manuscript, Ahmad Salti et al described an enhanced glycolytic activity mediated elevated iPSCs induction from somatic MEFs by FAHD1 functional loss. It’s a well-designed study with extensive discussion through the glycolytic and OXPHOS topic in terms of their function in the OSKM induction of iPSCs, as well as the attempt to reach a mechanistic explanation underlying this phenotype.

Response: We are grateful to the reviewer for these encouraging statements.

However, there are a few concerns that may still draw the author’s attention as listed below.

1. In the statement of elevated induction efficiency of iPSCs from FADH1-KO MEFs, it’s impressive that the authors ruled out the possibility of accelerated growth of MEFs brought by FADH1 knockout. However, as far as I can tell, it’s very likely that the observed “elevated iPSC induction rate” can be also an artifact due to the enhanced growth rate of FADH1-KO iPSCs.

Response: We thank the reviewer for his comment. We clarify now this point in the discussion (Line 435-437). In fact, during reprogramming, we observe an increase in the number of iPSC-like colony formation in Fahd1-KO, each colony representing an independent iPSC clone. Even if some clones have an enhanced growth rate due to the reprogramming, we are counting the number of colonies formed regardless of the cell number. However, our observation also indicate no difference in cell proliferation nor neuronal differentiation in KO iPSCs as compared to the WT as indicated by microscopic observation and similar expression of pluripotency, neural progenitor and neuronal markers (Figures 2 and new Figure 5A). It has also been shown by several studies that increased glycolysis enhances stem cell reprogramming, in concordance of what we also observe in Fahd1-KO MEFs.

2. Even though it’s explicitly stated in the manuscript that “Fahd1 deficiency does not affect pluripotency of iPSCs after reprogramming nor their differentiation to neurons”. However, this gene is actually known for hereditary tyrosinemia and its loss would lead to severe liver and renal disease. Thus the translational potential of FADH1-KO or inhibition in iPSCs is limited unless a transient inhibition is feasible. Additional study or discussion of tyrosine accumulation in FADH1-KO iPSC and neuron differentiation should be added.

Response: Based on this comment, we concede that, whereas the most common pluripotency markers are expressed to a similar extent in iPSC of either genotype, we have not addressed all aspects of pluripotency in iPSC of WT vs. Fahd1-KO genotype. Accordingly, we toned down the statement in the revised manuscript (Lines 254 and 497). The reviewer is also concerned about the translational potential of FAHD1 inhibition. This is indeed an open question. However, FAHD1 is not participating in tyrosine breakdown in eukaryotes; instead, this is the function of fumarylacetoacetate hydrolase (FAH), the founding member of the FAH superfamily which hydrolyses fumarylacetoacetate (FAA) in the cytosol; accordingly, FAH deficiency causes hereditary tyrosinemia in humans. FAHD1, on the other hand, is a mitochondrial protein, is unable to cleave FAA, and plays a key role in fine-tuning energy metabolism in various mammalian tissues, including the kidney and liver. We apologize for not making this point clearer and added an appropriate statement to the revised manuscript (Line 55-57).

3. FADH1 encodes a mitochondrial enzyme with dual substrate specificity that both hydrolyze acylpyruvate and decarboxylate oxaloacetate. However, the discussion section seems never referring its function or connecting its function to the mitochondrial dysfunction, rather put them all in the a box as “mitochondrial dysfunction”. In depth discussion in this manner is required.

Response: We apologize for not providing in the original manuscript a detailed description how FAHD1 deficiency is linked to mitochondrial dysfunction. We added a paragraph to the discussion section, along with Supplementary Figure S3, in which the current knowledge about FAHD1 as a regulator of mitochondrial function is summarized (Line 420-425).

Reviewer 2 Report

In this manuscript, the authors investigate the consequences of FAHD1 knockout in MEFs, iPSCs reprogramming and neuronal differentiation. The manuscript is well written and the topic or research well introduced. However, some aspects require attention.

  1. Although the importance of FAHD1 in mitochondrial function has been previously studied by the authors, it is unclear whether this protein is (directly) involved in the processes investigated here. Next to the comparison WT vs. FAHD1 KO, it is relevant to investigate if FAHD1 transcription, protein levels and/or activity change during ‘basal’ generation of iPSCs or iPSC differentiation (germ layer and neuronal). This could strogner support the overall message.

  1. Related to figure 3. Non-glycolytic acidification is not a direct/straightforward measurement of the bioenergetics state. Related to that, glycolysis is calculated as: Glycolysis= (Maximum rate measurement before Oligomycin injection) – (Last rate measurement before Glucose injection). How were the bioenergetics parameters calculated? (Relevant guidelines: https://www.agilent.com/cs/library/usermanuals/public/Report%20Generator%20User%20Guide_Seahorse%20XF%20Glycolysis%20Stress%20Test_RevA.pdf )

  1. Unexpectedly, mitochondrial respiration is upregulated when comparing iPSCs to MEFs. Related to that, a reduction in mtDNA copy number has been reported in the induction of iPSCs. Comparing mtDNA copy number between MEFs and iPSCs is currently not possible, because of normalization to WT (Figures 3C and 4C). This should be corrected/added and the outcome discussed

  1. Related to figure 5. The authors should include controls to monitor for the efficiency of neuronal differentiation both in WT, and compare it to FAHD1 KO.

  1. Related to figure 5. To better understand the importance of FAHD1 in neuronal differentiation, and for cohesion with previous figures, the authors should include Seahorse analysis of WT and FAHD1 KO neuronal differentiation.

  1. In line with the previous point and referred to in line 483. Mitochondrial dysfunction in neuronal differentiated cells is not addressed neither proven.

  1. Line 483 needs attention. Mitochondria respiration in MEFs is partially reduced in FADH1 KO, however it does not correlated with an increase in mtDNA copy number (Fig. 3 B and C). In iPSCs, the differences in respiration between WT and FAHD1 are arguable (4B and E), and in this case increase mtDNA copy number is observed. Accumulation of mitochondria can also result from defective mitophagy/degradation. This should be investigated and at least it should be discussed.

Minor:

  1. The authors should consider adding control results showing the knockout of FAHD1 in their system.
  2. The discussion is long and could benefit from editing. Rather than rewriting the outcome of the results, it should integrate the outcomes of the different processes studied here. Next to that, the main findings should be contextualized within the current understanding of the research area. Importantly, there is no discussion about the molecular function of the FAHD, as how (mechanistically) the knockout of the protein leads to the observed phenotypes.

Author Response

In this manuscript, the authors investigate the consequences of FAHD1 knockout in MEFs, iPSCs reprogramming and neuronal differentiation. The manuscript is well written and the topic or research well introduced. However, some aspects require attention.

Response: We are grateful to the reviewer for these encouraging statements.

1. Although the importance of FAHD1 in mitochondrial function has been previously studied by the authors, it is unclear whether this protein is (directly) involved in the processes investigated here. Next to the comparison WT vs. FAHD1 KO, it is relevant to investigate if FAHD1 transcription, protein levels and/or activity change during ‘basal’ generation of iPSCs or iPSC differentiation (germ layer and neuronal). This could strogner support the overall message.

Response: We are grateful to the reviewer for this suggestion. According to the request, data on the expression of Fahd1 in WT MEFs, iPSCs and neurons was added to the revised manuscript (Lines 259-261 and 393-398). As expected, expression of FAHD1 was not detectable in any of the FAHD1-KO cell types, and this is corroborated by Western blot analysis. Of particular interest in light of this reviewer comment, analysis of FAHD1 gene expression in WT cells revealed high expression in MEF, transient downregulation during reprogramming and strong upregulation during neuronal differentiation, suggesting a direct involvement of FAHD1 in the transition between these three states. These new results are provided in Supplementary Figure S2 C and D.

2. Related to figure 3. Non-glycolytic acidification is not a direct/straightforward measurement of the bioenergetics state. Related to that, glycolysis is calculated as: Glycolysis= (Maximum rate measurement before Oligomycin injection) – (Last rate measurement before Glucose injection). How were the bioenergetics parameters calculated? (Relevant guidelines: https://www.agilent.com/cs/library/usermanuals/public/Report%20Generator%20User%20Guide_Seahorse%20XF%20Glycolysis%20Stress%20Test_RevA.pdf )

Response: We thank the reviewer for providing us the link to the seahorse report generator and apologize that we had not clearly distinguished between which parameters were calculated and which one was measured. We have added a sentence for clarification in the results (Line 280-283) and the method section (Line 205-2011). We would like to highlight that the calculations and the measurement were automatically carried out by the Seahorse Report Generator (a Microsoft Excel Macro that automatically calculates and reports assay parameters of the Seahorse XF Mitochondrial and Glycolysis stress tests).

3. Unexpectedly, mitochondrial respiration is upregulated when comparing iPSCs to MEFs. Related to that, a reduction in mtDNA copy number has been reported in the induction of iPSCs. Comparing mtDNA copy number between MEFs and iPSCs is currently not possible, because of normalization to WT (Figures 3C and 4C). This should be corrected/added and the outcome discussed

Response: As requested by the reviewer, we added data on the regulation of mitochondrial DNA copy number between MEFs and iPSCs to the revised manuscript. These results suggest , in concordance with the literature,  a reduction in mt DNA copy number in WT cells during reprogramming, whereas mt DNA copy number is upregulated with reprogramming of FAHD1 KO cells. These results are now explained in the results (Line 345-353) and presented as Supplementary Figure S2 A and B.

4. Related to figure 5. The authors should include controls to monitor for the efficiency of neuronal differentiation both in WT, and compare it to FAHD1 KO.

Response: As requested by the reviewer, we added to the revised manuscript controls to monitor for the efficiency of neuronal differentiation by immunocytochemistry. This result is provided as new Figure 5A.

5. Related to figure 5. To better understand the importance of FAHD1 in neuronal differentiation, and for cohesion with previous figures, the authors should include Seahorse analysis of WT and FAHD1 KO neuronal differentiation.

Response: We agree with the reviewer that Seahorse analysis of neurons would be of general interest but consider this data beyond the scope of the current manuscript which is focused on the role of FAHD1 in the reprogramming process per se. The precise role of FAHD1 in neuronal differentiation will be further addressed in future studies. A statement in this regard was added to the discussion (Line 527).

6. In line with the previous point and referred to in line 483. Mitochondrial dysfunction in neuronal differentiated cells is not addressed neither proven.

Response: We agree with the reviewer’s comment that mitochondrial dysfunction in neurons has not been proven by the data shown in the current manuscript. We agree that such data would be of general interest but consider this data beyond the scope of the current which is focused on the role of FAHD1 in the reprogramming process per se. The precise role of FAHD1 in iPSC-derived neurons will be further addressed in future studies. A statement in this regard was added to the discussion (Line 527).

7. Line 483 needs attention. Mitochondria respiration in MEFs is partially reduced in FADH1 KO, however it does not correlated with an increase in mtDNA copy number (Fig. 3 B and C). In iPSCs, the differences in respiration between WT and FAHD1 are arguable (4B and E), and in this case increase mtDNA copy number is observed. Accumulation of mitochondria can also result from defective mitophagy/degradation. This should be investigated and at least it should be discussed.

Response: As requested by the reviewer, we added two sentences in the discussion section concerning conclusions on the observations of mitochondria DNA copy number changes (Line 494-496).

 Minor:

1. The authors should consider adding control results showing the knockout of FAHD1 in their system.

Response: As requested by the reviewer, we included in the revised manuscript data from RT-qPCR and Western blot (Lines 259-261) clearly demonstrating the KO phenotype (Supplementary Figure S2C and D), in agreement with genotyping results for the mouse strain from which the MEF were derived (not shown).

2. The discussion is long and could benefit from editing. Rather than rewriting the outcome of the results, it should integrate the outcomes of the different processes studied here. Next to that, the main findings should be contextualized within the current understanding of the research area. Importantly, there is no discussion about the molecular function of the FAHD, as how (mechanistically) the knockout of the protein leads to the observed phenotypes.

Response: In response to the reviewer’s comment, we edited the discussion section including a more detailed explanation of our hypothesis concerning the mechanism by which FAHD1 supports mitochondrial function (Line 420-425). We also included a graphical representation of this process (new Supplementary Figure S3) and referred the reader to a review article published in MAD recently (ref. 9).

Reviewer 3 Report

This is an interesting paper that studies role of FAH domain containing protein 1 (FAHD1) in the reprogramming of murine embryonic fibroblasts into iPSCs and iPSC differentiation into neuronal cells.Data indicates that FAHD1 plays a significant role in reprogramming efficiency and is responsible to the glycolytic activity attenuation. Based on the study findings, authors conclude that “FAHD1 as a potential negative regulator of reprogramming”.

My main concern is that all observations are discussed with a limited scope.

-It will add more clarity if authors include a schematic presentation of FAHD1 metabolized pathways and how they are connected to Krebs cycle.

- What is the etiology of increase in glycolysis in Fahd1-KO MEF? Is glucose uptake also increased? Did authors analyze expression of glucose transporters in Fahd1-KO MEF?

-How ATP levels change in Fahd1-KO MEF or expected to be changed? This need to be discussed.

-There is no change in basal respiration (Fig. 3B)- This need to be discussed.

- Analysis in neurons was performed 20 days post differentiation. In vivo, the metabolic surrounding also includes fatty acids and other carbon substrates.  How this is expected to change observations/data? It is possible that the presence of fatty acids will change dramatically mitochondrial biogenesis. If these experiments were not performed, it needs to be at least discussed

Author Response

This is an interesting paper that studies role of FAH domain containing protein 1 (FAHD1) in the reprogramming of murine embryonic fibroblasts into iPSCs and iPSC differentiation into neuronal cells. Data indicates that FAHD1 plays a significant role in reprogramming efficiency and is responsible to the glycolytic activity attenuation. Based on the study findings, authors conclude that “FAHD1 as a potential negative regulator of reprogramming”.

Response: We are grateful to the reviewer for these encouraging statements. 

My main concern is that all observations are discussed with a limited scope.

Response: The reviewer is concerned about the limited scope of our discussion section which is in part due to the fact that research on FAHD1 is still in its infancy. Nevertheless, we think that the current knowledge of FAHD1 function is appropriate to provide a rationale for the observations we report in this communication. We apologize for not being more precise in the original manuscript and have tried to improve the discussion of our findings with more explicit links to our published biological studies and the corresponding working hypothesis (Line 420-425).

-It will add more clarity if authors include a schematic presentation of FAHD1 metabolized pathways and how they are connected to Krebs cycle.

Response: As suggested by the reviewer, we included a schematic presentation of FAHD1’s role in energy metabolism and its connection to the Krebs cycle in Supplementary Figure S3.

- What is the etiology of increase in glycolysis in Fahd1-KO MEF? Is glucose uptake also increased? Did authors analyze expression of glucose transporters in Fahd1-KO MEF?

Response: It is well established in the field that inhibition of mitochondrial oxidative phosphorylation, by various different manipulations, leads to an increase in glycolysis in most cells, in order to compensate for reduced mitochondrial ATP production. We included in the revised manuscript a recent review article discussing this link (Line 448-451). Whether in our experimental system this process is accompanied by an increase in glucose uptake in FAHD1-KO MEFs remains to be determined and is considered beyond the scope of the current manuscript.

-How ATP levels change in Fahd1-KO MEF or expected to be changed? This need to be discussed.

Response: According to our hypothesis and supported by published and unpublished experimental data, the current view is that FAHD1 KO leads to a decreased activity of complex II of the mitochondrial electron transport chain, resulting from accumulation of oxaloacetate (OAA) which is a competitive inhibitor of succinate dehydrogenase (SDH). This working hypothesis is displayed in Supplementary Figure 3 (Line 420-425). OAA accumulation will lead to reduced mitochondrial ATP production accompanied by reduced flux through the TCA cycle which is most likely compensated for by increased glycolysis.

-There is no change in basal respiration (Fig. 3B)- This need to be discussed.

Response: The reviewer correctly notes that there is no change in basal respiration between WT and FAHD1 KO iPSC. We have added the sentence into the discussion section (Line 445-451) to explain this surprising finding.

- Analysis in neurons was performed 20 days post differentiation. In vivo, the metabolic surrounding also includes fatty acids and other carbon substrates.  How this is expected to change observations/data? It is possible that the presence of fatty acids will change dramatically mitochondrial biogenesis. If these experiments were not performed, it needs to be at least discussed

Response: The reviewer is right by stating that the analysis of neurons resulting from the differentiation of a PSC is not complete. We agree with the reviewer that the metabolic surrounding of neurons in vivo is different from cell culture conditions and for sure includes fatty acids and other carbon substrates. We think that a detailed analysis of these questions is beyond the scope of the current manuscript, which  provides for the first time the finding that FAHD1 plays in important role as modulator of cellular reprogramming. According to the reviewer’s request, we included a short paragraph in the discussion section of the revised manuscript (Line 518-522) where we discussed this point.

Round 2

Reviewer 2 Report

The additions have improved the manuscript. 

In the discussion new information has been added regarding the function of FADH1 on mitochondrial metabolism (lines 420- 425). A new figure/scheme has been now included (Fig. S3). Although the scheme is clear, the authors should better integrate this information within the findings of the current study. How would the proposed mechanisms connect/explain the phenotypes measured here? (mtDNA copy number, mito. respiration, etc). Referring to that briefly and adding, for instance, dashed arrows to the scheme can improve this aspect. 

Author Response

The additions have improved the manuscript. 

Response: We are grateful for the reviewer who made important comments that made the manuscript more clear and improved.

In the discussion new information has been added regarding the function of FADH1 on mitochondrial metabolism (lines 420- 425). A new figure/scheme has been now included (Fig. S3). Although the scheme is clear, the authors should better integrate this information within the findings of the current study. How would the proposed mechanisms connect/explain the phenotypes measured here? (mtDNA copy number, mito. respiration, etc). Referring to that briefly and adding, for instance, dashed arrows to the scheme can improve this aspect. 

Response: Thank you for the valuable comment. We agree with the reviewer comment and made the necessary corrections accordingly. We now updated the previously (Fig. S3) and its figure legend to show how the proposed mechanisms connect with the phenotypes measured. We also included now the Figure in the main manuscript text as new Figure 6. We updated the text in the discussion and moved it to the end as a proposed conclusion (Line 538-546).

Reviewer 3 Report

Authors addressed all my comments/concerns. I recommend to accept the manuscript for the publication.

Author Response

Authors addressed all my comments/concerns. I recommend to accept the manuscript for the publication.

We are grateful for the valuable comments of the reviewer, which made the manuscript more clear and improved.